# Effects of Neurofeedback in Children with Attention-Deficit/Hyperactivity Disorder: A Systematic Review

**DOI:** 10.3390/jcm10173797

**Published:** 2021-08-25

**Authors:** Lucía Sampedro Baena, Guillermo A. Cañadas-De la Fuente, María Begoña Martos-Cabrera, José L. Gómez-Urquiza, Luis Albendín-García, José Luis Romero-Bejar, Nora Suleiman-Martos

**Affiliations:** 1San Cecilio Clinical University Hospital, Andalusian Health Service, Avenida del Conocimiento, s/n, 18016 Granada, Spain; luciabaena98@hotmail.com (L.S.B.); bego_martos@hotmail.com (M.B.M.-C.); 2Faculty of Health Sciences, University of Granada, Avenida de la Ilustración, 60, 18016 Granada, Spain; gacf@ugr.es; 3Faculty of Health Sciences, University of Granada, Calle Cortadura del Valle S.N., 51001 Ceuta, Spain; jlgurquiza@ugr.es (J.L.G.-U.); norasm@ugr.es (N.S.-M.); 4Casería de Montijo Health Center, Granada Metropolitan District, Andalusian Health Service, Calle Virgen de la Consolación, 12, 18015 Granada, Spain; lualbgar1979@ugr.es; 5Department of Statistics and Operational Research, University of Granada. Av. Fuentenueva, 18071 Granada, Spain

**Keywords:** attention deficit/hyperactivity disorder, child, neurofeedback, treatment

## Abstract

Attention deficit/hyperactivity disorder (ADHD) is one of the most frequent neurodevelopmental disorders in childhood and adolescence. Choosing the right treatment is critical to controlling and improving symptoms. An innovative ADHD treatment is neurofeedback (NF) that trains participants to self-regulate brain activity. The aim of the study was to analyze the effects of NF interventions in children with ADHD. A systematic review was carried out in the CINAHL, Medline (PubMed), Proquest, and Scopus databases, following the PRISMA recommendations. Nine articles were found. The NF improved behavior, allowed greater control of impulsivity, and increased sustained attention. In addition, it improved motor control, bimanual coordination and was associated with a reduction in theta waves. NF combined with other interventions such as medication, physical activity, behavioral therapy training, or attention training with brain–computer interaction, reduced primary ADHD symptoms. Furthermore, more randomized controlled trials would be necessary to determine the significant effects.

## 1. Introduction

Attention deficit/hyperactivity disorder (ADHD) or hyperkinetic disorder is one of the most common mental disorders in childhood and adolescence [1]. It is a neurodevelopmental disorder characterized by primary symptoms of inattention, hyperactivity and/or impulsivity [2]. There are three types of ADHD, inattentive, hyperactive-impulsive, and combined type [3]. Worldwide, its prevalence is 5.8% [2,4].

There are different genetic and environmental factors among the causes of attention deficit/hyperactivity disorder. ADHD tends to run in families, with a risk of 5 to 10 times higher among first-degree relatives [5]. Regarding environmental factors, these mainly include prenatal and perinatal risk factors (maternal stress, smoking or alcohol consumption during pregnancy, low birth weight, prematurity), environmental toxins (organophosphates, polychlorinated biphenyls, lead) and unfavorable psychosocial conditions (maternal hostility, severe deprivation in early childhood) [5].

Throughout childhood, ADHD is related to inattention, poor planning ability, and impulsivity, causing further deterioration as external demands increase [6]. This generates a series of alterations in personal, school, and social functionality, which lead the individual in the full stage of formation of his personality and identity, to interact in an erroneous way with society, causing conflicts with the environment (parents, siblings, colleagues) and that fact can lead to social marginalization [7,8]. These children, from preschool to 13 years of age, show a risk of suicidal ideation almost six times greater than that of a child without ADHD [2,9].

Therefore, early interventions in children are essential, in order to reduce the repercussions in adolescence and adulthood. Among the effective treatments available for ADHD, the main differences are related to the type of intervention (pharmacological and non-pharmacological), the age of the patient, the cost, the available patient and caregiver time, the expected effectiveness in the reduction of symptoms, the adverse effects, safety, and tolerability [10,11].

The main treatment for ADHD continues to be pharmacological, with psychostimulant drugs (methylphenidate or amphetamines) being the most widely used [12]. However, the benefits are limited due to frequent adverse effects such as decreased appetite, headache, and insomnia, as well as poor adherence to treatment [5]. This fact makes many families reject the medication [13].

Among the different non-pharmacological treatment strategies, neurofeedback (NF) has been considered an innovative ADHD treatment [14]. NF is a computer-based behavior training enabling a patient to self-regulate aspects of brain activity [15]. The training protocols followed in ADHD procedures are training of slow cortical potentials (SCPs), theta/beta wave training and sensory-motor rhythm training (SMR) [16].

The NF focuses mainly on improving self-control over patterns of brain activity, decreasing theta waves (low frequency waves related to decreased alertness), and/or increasing beta waves (high frequency waves related to concentration and neuronal excitability) [17,18]. This is achieved by measuring the activity of the electroencephalography (EEG) while the patient performs a task, often a simple computer game, that modulates performance and reward according to specific changes in the EEG pattern [19].

Some systematic reviews have analyzed the positive effects of non-pharmacological interventions [20,21], and other studies have compared the effects of NF interventions versus psychostimulants [22]. Also, in the last years, a meta-analysis in children and adolescents suggested sustained symptom reductions over time after NF interventions [16]. However, the age range of the study population was large and the authors indicated serious limitations related to the scarcity of studies found and short-terms follow-up. Recently, an increasing number of studies investigating NF interventions with longer follow-up have been published, so researching the latest evidence can provide more information about this population. Therefore, the aim of this systematic review was to analyze the effects of interventions with NF exclusively in children with ADHD. 

## 2. Materials and Methods

### 2.1. Design and Search Strategy

A systematic review was performed following the guidelines of the PRISMA statement (Preferred Reporting Items for Systematic Reviews and Meta-Analyses) [23].

The search was carried out through the following databases: CINAHL, Medline (via PubMed), Proquest, and Scopus. The search equation using the Mesh terms was “Child AND attention deficit hyperactivity disorder AND neurofeedback”. The search was conducted in June 2021.

### 2.2. Inclusion and Exclusion Criteria

The inclusion criteria were: (1) randomized controlled trials, (2) children sample (up to 13 years), (3) NF as a treatment for ADHD, (4) comparison of NF with a controlled group or other interventions, (5) published in English or Spanish, (6) published in the last 5 years.

The exclusion criteria were: (1) no randomization, (2) mixed samples (adults and children) or adult population sample, (3) interventions without NF, (4) only NF interventions without a control group or other interventions.

### 2.3. Study Selection, Quality Appraisal, and Risk of Bias

The selection of articles was carried out in three steps. The title and abstract were read first, followed by reading the full text. Finally, a critical reading of the studies was carried out to assess the methodological quality and risk of bias.

The quality of the included studies was assessed following the levels of evidence and grades of recommendation by the Oxford Center for Evidence-Based Medicine (OCEBM) [24]. Risk of bias was analyzed by pairs of independent reviewers following the standards in the Cochrane Collaboration Risk of Bias tool for clinical trials [25].

### 2.4. Data Abstraction

First, two authors independently reviewed the title and abstract of the articles found. Second full text was read. A third author was consulted in case of disagreement.

To extract the data from each study, a data collection table was created. The variables obtained in the selected articles were: (1) author, year of publication, country; (2) design; (3) sample size; (4) aim; (5) type of intervention, duration, and follow-up; (6) main results.

## 3. Results

### 3.1. Study Characteristics

The initial search provided 332 articles. After removing duplicates and those that did not meet the inclusion criteria, a total of *n* = 9 studies were included. The study selection process is shown in Figure 1.

All included studies were randomized controlled trials [26,27,28,29,30,31,32,33,34]. The total sample was 620 children. Most of the studies were conducted in Iran (*n* = 3). The main characteristics of all the included studies are listed in Table 1.

### 3.2. Training through NF: Slow Cortical Potentials (SCP), Sensory-Motor Rhythm Training (SMR), and Theta-Beta Waves

Comparing the clinical efficacy of the NF-SCP and the computerized cognitive training (CogT), an improvement was found in the symptomatology perceived by parents and teachers after training. Furthermore, the effect of NF-SMR combined with game-based CogT in children with ADHD showed positive therapeutic effects on brain waves, except for sensory motor activity (no significant changes in the waves of frontocentral zero) as well as an improvement in symptomatology [32].

On the other hand, it is found that NF-SMR training improved motor control and more specifically bimanual coordination among children with ADHD [31]. In addition, a reduction in theta waves in the right and left hemispheres was recorded in participants with NF condition (t (17) = 3.73, *p* < 0.01 for the left hemisphere and t (17) = 3.97, *p* < 0.01 for right hemisphere) [28], although it was not recommended for the use of theta/beta training as an independent and exclusive treatment to improve neurocognitive functioning in children with ADHD [27].

### 3.3. Neurofeedback vs. Drug Treatment

Comparing NF with methylphenidate (MPH) treatment, teachers reported significantly lower ADHD symptoms in the group MPH, but there were no differences between groups in parental report [34]. In addition, it was observed that the combined treatment NF and MPH, improved ADHD symptoms (*p* = 0.01), being more effective compared to single medication treatment [26]. Some authors point out that NF improved attention, hyperactivity, and impulsivity while drug treatment improved visual attention capacity [30]. Although, on the other hand, other authors did not find superior effects of combined NF on intelligent functioning compared to the medication treatment only [28], and others found that stimulant medication showed superior effects over NF to improve neurocognitive functioning [27].

### 3.4. Neurofeedback vs. Other Interventions

Some papers compared the effects of NF with a physical activity intervention [27,33]. Combining both methods, NF plus physical activity, improved attention and short/long-term memory, although NF seem to lead to better and broader progress [33]. Other authors did not find significant benefits [27].

Analyzing NF and the behavioral therapy intervention, some authors found that both treatments achieved improvement on response control and attention [30].

Finally, comparing the NF with attention training based on the brain–computer interface, an improvement was found in inattentive symptoms in ADHD children, being an option for treating milder cases or as an adjunctive treatment (difference between groups of 1.6; 95% CI 0.3 to 2.9 *p* = 0.0177) [29].

## 4. Discussion

ADHD is one of the most common neurological disorders in childhood. Although there are different therapeutic interventions, the goal of this research was to evaluate the efficacy of NF interventions in children.

NF improved the symptomatology in children with ADHD, because of learning through video games, attending to their mistakes, and training functions nonconscious control—such as attention, achievement of objectives, self-control, and self-regulation of attention levels, and concentration—in addition to inhibiting distracting stimuli [35]. Also, it was found evidence of comparative effectiveness of NF and CogT for children with ADHD. As other studies point out, NF is a treatment with great benefits due to its positive and lasting effect on symptoms [3], significantly improving behaviour, attention, IQ [36,37], and reaction times on the psychometric measures such as hyperactivity and impulsivity symptoms [37].

In addition, another of the benefits found was the improvement in motor control and bimanual coordination. ADHD in children is often associated with poor motor control, coordination problems, and difficulty controlling strength [38], which affects the performance of distal, complex, and accelerated tasks that lead to poor writing and poor academic performance [39].

It seems that the NF is related to a reduction in theta waves, although the results were not clear. Literature also shows divergences, some authors found significant individual learning curves for both theta and beta over the course of the intervention, although individual learning curves were not significantly correlated with behavioral changes [37,40]. Other authors provided evidence that children with ADHD learned to decrease theta/beta ratio during NF sessions being the learning effects mainly attributable to the increase in the power of the beta waves both, in the group level sessions and in the individual level sessions. However, it is unknown whether the decrease in theta waves is attributed to the efficacy of NF sessions or to developmental changes in children [40].

Regarding drug treatment, NF is considered a viable alternative therapy for specific groups of children with ADHD who do not respond to medication or have severe side effects [5,22]. The combined use of drugs and NF enhances the durability of the positive effects [17] and even other studies corroborate that multimodal treatment of NF plus stimulant medication at low doses was positive in improving symptoms such as inattention and hyperactivity [22]. As other studies point out, drug doses can be reduced when both treatments are administered [3,41].

Alternative therapies combining NF plus physical activity improved attention and short/long-term memory. Other studies indicate great benefits in the improvement of cognitive, behavioral and physical symptoms after mixed physical exercise programs, particularly moderate to intense aerobic exercise [42,43]. Indeed, physical exercise generates an increase in the levels of norepinephrine, serotonin, and dopamine [44], which translates into the better motor and cognitive functioning and greater control of executive functions and impulses, so many authors bet on complementing NF and physical activity [45].

Behavioral therapy with NF improved response and attention control, as pointed out by other authors [46]. Even other studies indicate that NF through video game-based cognitive training normalizes brain function in patients with ADHD, since these therapeutic interventions are generally effective in improving cognitive areas and producing a decrease in symptoms of ADHD [47].

NF combinate with the training of attention based on the brain–computer interface shows benefits in inattentive symptoms, even some authors indicate that this type of training program is beneficial in cases of ADHD that present more severe symptoms [48].

One promising aspect of NF is its relationship to procedural learning, so its benefits can be more durable over time after treatment. Technology development provides different treatment options, combining traditional drug treatment, along with NF, physical activity, or cognitive training, all strategies provide positive therapeutic effects on brainwaves and ADHD symptomatology. 

Although NF can be used as complementary in patients who have significant side effects to stimulant medication or in patients whose family refuses to try the medication, further research is needed to corroborate its effects in each of the behavioural areas of the children with ADHD.

### Limitations

This study had a number of limitations to be acknowledged. First, although all studies use NF as an intervention, the great variability in the duration of the intervention may influence the heterogeneity of the results. Also, the sample sizes of the controlled randomized trial were small. Future research would be necessary to analyze the NF effects in each of the behavior areas of children with ADHD. Furthermore, more randomized controlled trials would be necessary to determine the significant effects and the duration of the effects over time.

## 5. Conclusions

NF showed a significant improvement of the symptoms in children with ADHD. This treatment improved behavior, allowed greater control of impulsivity, and increased sustained attention. In addition, it improved motor control, bimanual coordination, and was associated with a reduction in theta waves. NF combined with other interventions such as drugs, physical activity, behavioral therapy training, or attention training with brain–computer reduced primary ADHD symptoms. However, more scientific evidence is needed to support its efficacy and greater implantation in clinical practice.

## Figures and Tables

**Figure 1 jcm-10-03797-f001:**
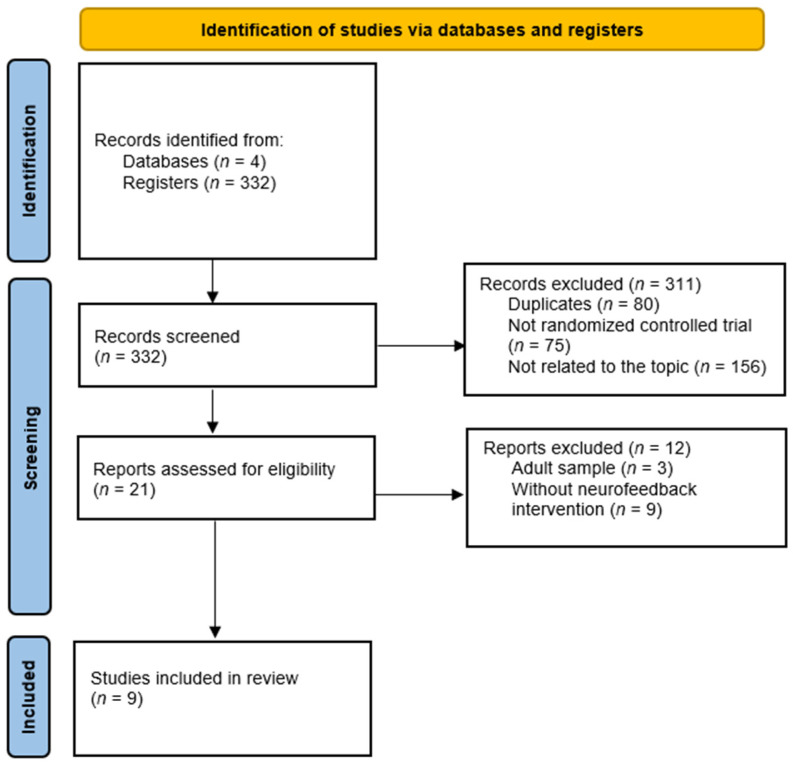
PRISMA 2020 flow diagram.

**Table 1 jcm-10-03797-t001:** Characteristics of the included studies (*n* = 9).

Authors, (Year), Country	Design	Sample	Aim	Measuring Duration/Follow-Up	Main Results Baseline/Follow-Up	EL/RG
Duric et al., [26], 2017, Norway	RCT	*n* = 130 Mean age = 10.9 (3.35) years G1 medication *n* = 44 G2 medication + NF *n* = 44 G3 NF *n* = 42	To explore efficacy of medication and NF	Disruptive Behaviour Disorders Rating Scale (Barkley’s Manual): - Parent Form - Teacher Form - Clinical Interview–Parent Report Form 30 sessions 6 months	**Disruptive Behaviour Disorders Rating Scale** Mean (95% CI) Teacher *Attention* G1: 17.8 (16.3–19.3)/13.1 (11.0–15.1) G2: 19.4 (17.8–21.1)/11.6 (9.3–13.8) G3: 15.6 (14.1–17.1)/14.8 (12.7–16.8) *Hyperactivity* G1: 11.6 (9.1–14.1)/13.0 (10.7–15.3) G2: 8.6 (5.8–11.4)/10.5 (8.0–13.0) G3: 11.6 (9.0–14.2)/10.5 (8.2–12.8) Parents *Attention* G1: 17.8 (16.0–19.5)/12.1 (10.2–14.0) G2: 17.2 (15.4–19.0)/11.8 (9.8–13.7) G3: 15.3 (13.4–17.2)/13.9 (11.9–15.9) *Hyperactivity* G1: 18.5 (16.5–20.5)/11.4 (9.4–13.5) G2: 15.9 (13.9–17.9)/10.9 (8.8–13.0) G3: 16.8 (14.8–18.9)/10.0 (7.8–12.1) Self-report children *Attention* G1: 5.4 (4.5–6.3)/6.0 (5.2–6.8) G2: 5.2 (4.2–6.1)/6.9 (6.1–7.7) G3: 4.7 (3.7–5.7)/5.6 (4.8–6.5) *Hyperactivity* G1: 5.2 (4.2–6.1)/5.9 (5.0–6.8) G2: 6.0 (5.1–7.0)/5.9 (5.0–6.8) G3: 4.3 (3.3–5.3)/5.8 (4.8–6.7) *Education* G1: 5.2 (4.2–6.1)/5.9 (5.0–6.8) G2: 6.0 (5.1–7.0)/5.9 (5.0–6.8) G3: 4.3 (3.3–5.3)/5.8 (4.8–6.7)	1b/A
Geladé et al. [27], 2017, Netherlands	RCT	*n* = 112 Age 7–13 yearsG1 NF *n* = 39 G2 medication *n* = 36 G3 PA *n* = 37	To compare NF effects on neurocognitive functioning	Oddball task Stop-signal task Visual spatial working memory task 30 sessions 6–9 months	**Oddball task** M (SD) *Mean reaction time* G1: 440.72 (100.00)/433.54 (95.63) Adjusted difference [95%CI] −7.20 [−25.64, 11.30] G2: 461.56 (68.65)/404.40 (63.00) Adjusted difference [95%CI] −57.19 [−81.60, −32.80] G3: 438.01 (88.51)/447.02 (90.06) Adjusted difference [95%CI] 9.01 [−9.33, 27.35] **Stop-signal task** *Stop-signal reaction time* G1: 271.39 (76.00)/252.89 (83.60) Adjusted difference [95%CI] −19.03 [−38.42, 0.36] G2: 278.10 (91.40)/202.30 (96.20) Adjusted difference [95%CI] −94.92 [−123.90, −65.94] G3: 245.56 (84.83)/236.54 (84.06) Adjusted difference [95%CI] −9.02 [−29.79, 11.74] *Mean reaction time* Adjusted difference [95%CI] −32.40 [−49.51, −15.28] G1: 642.68 (123.71)/610.12 (122.39) G2: 679.78 (122.31)/629.35 (136.23) G3: 631.73 (110.42)/617.52 (123.86) **Visual spatial working memory** *Forward* Adjusted difference [95%CI] 0.71 [0.24, 1.17] G1: 12.26 (2.92)/12.67 (3.60) G2: 11.00 (2.58)/12.17 (2.72) G3: 11.16 (2.73)/11.68 (3.53) *Backward* Adjusted difference [95%CI] 1.32 [0.78, 1.86] G1: 10.90 (3.08)/11.67 (3.40) G2: 9.58 (2.50)/11.33 (3.60) G3: 9.95 (2.95)/11.00 (3.32)	1b/A
Lee & Jung [28], 2017, Korea	RCT	*n* = 36 Age 7–12 yearsCG medication *n* = 18 IG NF *n* = 18	To examine the effect of NF on cognitive functions, parental symptom reports, and brainwave activity before and after treatment	ADHD diagnostic system (ADS)ADHD Rating Scale (ARS) Conners Rating Scale Revised (CRS) Korean-Wechsler Intelligence Scale for Children-III (K-WISC-III) 20 twice-weekly sessions 2,5 months	**ADS** M(SD) *Inattention* CG: 107.22 (83.03)/66.00 (48.49) * IG: 78.39 (30.67)/52.67 (11.78) ** *Impulsivity* CG: 72.72 (27.22)/59.50 (20.70) * IG: 79.39 (36.21)/59.44 (17.87) * *Response time* CG: 53.61 (14.36)/51.67 (11.62) IG: 52.33 (9.25)/47.33 (11.24) *Variability* CG: 85.61 (38.25)/67.72 (25.67) IG: 85.61 (38.25)/67.72 (25.67) * **ARS** CG: 15.94 (2.24)/15.22 (2.86) IG: 14.33 (3.40)/10.78 (4.91) * **CRS** CG: 15.83 (6.71)/11.33 (5.03) ** IG: 13.89 (7.61)/7.61 (4.90) ** **K-WISC-III** *Full scale intelligence* CG: 100.72 (12.06)/110.72 (12.80) *** IG: 100.06 (16.60)/107.33 (16.93) *** *Verbal intelligence* CG: 100.44 (10.65)/105.28 (12.44) ** IG: 100.39 (16.21)/107.06 (15.39) * *Performance intelligence* CG: 101.06 (14.58)/114.61 (13.45) *** IG: 99.22(16.21)/107.06 (16.54) *** *Verbal comprehension* CG 101.94 (10.71)/105.61 (10.78) IG: 100.22 (15.89)/107.22 (15.40) *** *Perceptual organization* CG: 100.17 (15.51)/112.78 (15.19) *** IG: 101.67 (16.83)/109.78 (16.92) ** *Freedom from distractibility* CG: 93.94 (17.55)/100.39 (17.42) * IG: 95.94 (15.14)/100.39 (16.25) *Perceptual speed* CG: 99.61 (11.67)/105.06 (19.83) IG: 95.11 (13.56)/101.78 (12.36) **	1b/A
Lim et al. [29], 2019, Singapore	RCT	*n* = 172 Age 6–12 yearsCG *n*= 87 IG NF-BCI *n* = 85	To investigate the efficacy of NF attention training program	ADHD Rating Scale (ARS) 6 sessions 8-week	**ARS** M (SD) *Inattention score* MD (95% CI) 1.6 (0.3–2.9) *p* = 0.017 CG: 18.6 (4.38)/16.7 (5.14) IG: 18.9 (4.25)/15.5 (4.48)	1b/A
Moreno-García et al. [30], 2019, Spain	RCT	*n* = 57 Age 7–10 yearsG1 NF *n*= 19 G2 medication *n*= 19 G3 behavioural therapy *n*= 19	To examine the efficacy of NF on the improvement of symptoms	ADHD Rating Scales (ARS) Attention Deficit Disorders Evaluation Scale (ADDES) 40 sessions 20 weeks	**ARS** M (SD) Parents *Hyperactivity* G1: 17.43 (4.98)/14.21 (6.77) G2: 12.40 (8.69)/8.80 (5.82) G3: 15.38 (6.66)/9.94 (5.42) ** *Inattention* G1: 17.64 (4.63)/15.86 (6.81) G2: 19.75 (5.07)/14.31 (6.41) G3: 19.30 (5.41)/14.40 (3.83) ** Teacher *Hyperactivity* G1: 17 (9.09)/10.86 (8.35) * G2: 9.83 (7.88)/6.83 (4.57) G3: 12.10 (7.53)/12.10 (7.53) ** *Inattention* G1: 20.43 (4.89) /14.14 (5.30) G2:17.50 (8.09)/15.67 (4.03) G3: 13.90 (5.19)/13.90 (5.19) ** **ADDES** *Hyperactivity* G1: 47.38 (20.86)/38.63 (19.42) * G2: 33.55 (22.22)/21.00 (20.01) * G3: 44.94 (28.38)/30.63 (24.22) ** *Inattention* G1 54.25 (14.90)/45.63 (16.29) * G2 50.91 (14.43)/38.09 (21.61) * G3 55.88 (20.90)/36.25 (20.29) ***	1b/A
Norouzi et al., [31], 2018, Iran	RCT	*n* = 20 Age 6–10 years CG *n* =10 IG NF-SMR *n*= 20	To analyse NF for improving bimanual coordination	Bimanual coordination task 6−9 sessions 4 weeks	Bimanual coordination accuracy and consistency improved from baseline to follow up in NF group ***	1b/A
Rajabi et al., [32], 2020, Iran	RCT	*n* = 32 Mean age = 10.20 (1.03) years CG *n* = 16 IG NF + CogT *n*= 16	To examine the effects of NF	Integrated Visual and Auditory continuous performance (IVA) Conners Parent Rating Scales-Revised (CPRS-R) Conners Teacher Rating Scales-Revised (CTRS-R) 30 sessions 3 months	**IVA** M (SD) CG: 65.71 (16.8)/59.71 (11.1) IG: 58.20 (24.8)/80.50 (7.6) *** **CPRS-R** CG: 13.00 (3.5)/13.20 (3.3) IG: 11.91 (2.5)/9.61 (2.2) *** **CTRS-R** CG: 9.10 (3.9)/9.40 (3.8) IG: 11.11 (2.1)/7.21 (1.6) *	1b/A
Rezaei et al. [33], 2018, Iran	RCT	*n* = 21 Age 7–11 yearsCG *n* = 7 G1 NF *n* = 7 G2 yoga *n* = 7	To examine the effects of NF	Continues Performance Test (CPT) 24 sessions 8 weeks	**CPT** M (SD) *Correct Detection* CG: 30 (10.6)/33.1 (7.9) *** IG1: 16.71 (5.55)/4.71 (3.49) *** IG2: 21.71 (21.24)/9.57 (9.10) *** *Reaction time* CG: 577 (45)/590 (20) IG1: 611 (79)/555 (72) IG2: 591 (129)/587 (74) *Types of error* *Commission* CG: 107.7 (17.3)/105.1 (13) *** IG1: 130.9 (6.3)/144.1 (3.5) *** IG2: 118.1 (24)/135.1 (11) *** *Omission* CG: 12.3 (8.5)/11.7 (5.9) *** IG1: 2.42 (2.07)/1.14 (1.34) *** IG2: 10.14 (4.05)/5 (3.55) ***	1b/A
Sudnawa et al., [34], 2018, Thailand	RCT	*n* = 40 Mean age = 8.7 1.55) years CG medication *n* = 20 G1 NF *n* = 20	To evaluate the effectiveness of NF	Vanderbilt ADHD Diagnostic Rating Scales 30 sessions 12 weeks	**Vanderbilt scores** M (SD) Parent scores *Inattentive* CG: 17.1 (4.3)/10.1 (4.2) * IG: 15.3 (5.3)/11.8 (6.3) * *ADHD symptoms* CG: 33.1 (8.3)/20.1 (8.0) * IG: 29.1 (9.7)/22.0 (11.6) * Teacher scores *Inattentive* CG: 17.3 (4.2)/9.3 (6.1) * IG: 19.5 (3.1)/16.2 (6.3) * *ADHD symptoms:* CG: 32.9 (7.0)/16.0 (12.7) * IG: 34.0 (9.2)/27.6 (12.2)	1b/A

* *p* < 0.05; ** *p* < 0.01; *** *p* < 0.001. ADHD = Attention Deficit/Hyperactivity Disorder; BCI = brain computer interface -based attention training program; CG = Control group; CogT = Computerized Cognitive Training; EL= Evidence level; IG = Intervention group; NF = Neurofeedback; PA = physical activity; RG = Recommendation grade; SCP = Slow Cortical Potential; SMR = sensory-motor rhythm training.

## Data Availability

This paper is excluded due to not reporting any data.

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
