# Peer review of "Effects of Neurofeedback in Children with Attention-Deficit/Hyperactivity Disorder: A Systematic Review"

_jcm, 2021, doi:10.3390/jcm10173797_

Round 1
Reviewer 1 Report
The review article entitled "Effects of Neurofeedback in Children and Adolescents with Attention-Deficit/Hyperactivity Disorder: A Systematic Review" by Baena LS et al., summarizes the efficacy of neurofeedback interventions with different treatments in children and adolescents with ADHD.
The authors have collected, and summarized information relevant to the topic discussed. The authors have nicely presented the included studies in tabular forms. The topic of this review is relevant and of interest. The tabular presentations made the included studies quick to understand and compare. Though the review discusses some interesting point of the research area, it lacks in comprehensiveness and important insights from the authors themselves. The author has provided interesting topic to start with, but it would be great if they add some of their own critical input in the introduction/discussion portion. Other than this, the article is presented well and easy to follow. The review fits with the current interest and scopes of journal. I will recommend some further input from the authors in the introduction/discussion area which could certainly improve the overall scientific quality and acceptability of this review. Though the way the manuscript is presented, it could also be accepted based on the comments from other reviewers.
Author Response
Response to Reviewer 1 Comments
Dear Reviewer,
Thank you very much for reviewing the manuscript and your recommendations for improving it. Please find below the response to each recommendation highlighted in yellow. All the changes in the manuscript have also been highlighted in yellow.
The review article entitled "Effects of Neurofeedback in Children and Adolescents with Attention-Deficit/Hyperactivity Disorder: A Systematic Review" by Baena LS et al., summarizes the efficacy of neurofeedback interventions with different treatments in children and adolescents with ADHD.
The authors have collected, and summarized information relevant to the topic discussed. The authors have nicely presented the included studies in tabular forms. The topic of this review is relevant and of interest. The tabular presentations made the included studies quick to understand and compare. Though the review discusses some interesting point of the research area, it lacks in comprehensiveness and important insights from the authors themselves. The author has provided interesting topic to start with, but it would be great if they add some of their own critical input in the introduction/discussion portion. Other than this, the article is presented well and easy to follow. The review fits with the current interest and scopes of journal. I will recommend some further input from the authors in the introduction/discussion area which could certainly improve the overall scientific quality and acceptability of this review. Though the way the manuscript is presented, it could also be accepted based on the comments from other reviewers.
Dear Reviewer, thank you for your recommendations. As you have suggested, we have added more critical input in the introduction/discussion section.
Reviewer 2 Report
This systematic review focuses on determining whether neurofeedback is effective in reducing ADHD symptoms in children with ADHD. Such non-pharmacological treatments are important to assess, as many ADHD patients do not sufficiently respond to stimulant treatment, and there is also low adherence to medication regimes due to negative side effects. The systematic review is carried out in accordance with the PRISMA guidelines, and therefore has a robust methodology. However, one of my major concerns is the novelty of the results, as a systematic review and meta-analysis was also carried out on the effects of neurofeedback in childhood ADHD in 2019 by Van Doren et al. The authors therefore need to clarify how their systematic review is novel to this previous study, and what knowledge it adds, before their review can be considered for publication.
I also have several other minor comments, as listed below:
- There are multiple grammar mistakes, and several sentences do not flow or make sense. I therefore suggest that a native English speaker edits the manuscript.
- A mixture of English and American spelling is used – please select one and keep it consistent.
- The authors state several times that their review focuses on children and adolescents with ADHD, yet in the search criteria, they only included the term ‘child’. Therefore the results of the systematic review are limited to children.
- Pubmed was not included in the databases searched. What was the reasoning for this?
- The results section is very short and not very detailed. It would benefit from having some more data added, such as effect sizes and p-values.
- The author provide nice tables summarising each study included – however, no p-values are given. Please add these, to help readers interpret the results.
- In the discussion, the authors state that ADHD is related to a dopamine deficiency. This has not been conclusively proven, therefore I would word the sentence less strongly. Also, a reference for this is missing.
Author Response
Response to Reviewer 2 Comments
Dear Reviewer,
Thank you very much for reviewing the manuscript and your recommendations for improving it. Please find below the response to each recommendation highlighted in yellow. All the changes in the manuscript have also been highlighted in yellow.
Point 1. This systematic review focuses on determining whether neurofeedback is effective in reducing ADHD symptoms in children with ADHD. Such non-pharmacological treatments are important to assess, as many ADHD patients do not sufficiently respond to stimulant treatment, and there is also low adherence to medication regimes due to negative side effects. The systematic review is carried out in accordance with the PRISMA guidelines, and therefore has a robust methodology. However, one of my major concerns is the novelty of the results, as a systematic review and meta-analysis was also carried out on the effects of neurofeedback in childhood ADHD in 2019 by Van Doren et al. The authors therefore need to clarify how their systematic review is novel to this previous study, and what knowledge it adds, before their review can be considered for publication.
Response 1. We sincerely appreciate all valuable comments and suggestions. Van Doren et al. (2019), conducted a search for studies up to 2017, including children and young adults up to 18 years of age (age range that is too large that can lead to large biases). Also, their study indicates several limitations, between them, that “more carefully designed RCTs with longer follow-up time periods are needed before definite conclusions”, “This meta-analysis on follow-up effects at the present time may also allow us to derive the first relevant conclusions about the lasting effects of NF treatment”, and “largely missing from the literature that has conducted follow-up measurements”. Recently, an increasing number of RCTs only in children has been published (not included in their meta-analysis). In this sense, we performed this systematic review with the most recent articles published and only in children up to 13 years, which solves some indicated limitations, such as a longer follow-up time of the intervention with NF.
We have included more information Introduction section to clarify the differences.
I also have several other minor comments, as listed below:
Point 2.
- There are multiple grammar mistakes, and several sentences do not flow or make sense. I therefore suggest that a native English speaker edits the manuscript.
- A mixture of English and American spelling is used – please select one and keep it consistent.
Response 2. Thanks for the suggestion. Following the recommendations, we have sent our paper to an official translator, Glenn Harding. He has exhaustively checked the language and expressions used.
Point 3. The authors state several times that their review focuses on children and adolescents with ADHD, yet in the search criteria, they only included the term ‘child’. Therefore the results of the systematic review are limited to children.
Response 3. We only include studies with children up to 13 years of age. However, some studies considered their sample as children and adolescents. To avoid any confusion, we have eliminated the term adolescent. We have included the age of the sample in Table 1 and in the methods section.
Point 4. Pubmed was not included in the databases searched. What was the reasoning for this?
Response 4. PubMed is a search engine accessing primarily the MEDLINE database. To avoid any confusion, we have included that the search was done in Medline (via PubMed)
Point 5.
- The results section is very short and not very detailed. It would benefit from having some more data added, such as effect sizes and p-values.
- The author provides nice tables summarising each study included – however, no p-values are given. Please add these, to help readers interpret the results.
Response 5. We have included more statistical data (effect sizes and p-values) of those articles included that provide sufficient statistical information.
Point 6. In the discussion, the authors state that ADHD is related to a dopamine deficiency. This has not been conclusively proven, therefore I would word the sentence less strongly. Also, a reference for this is missing.
Response 6. Thank you for your appreciation. We have removed the sentence.
Round 2
Reviewer 2 Report
The manuscript is much improved and almost ready for publication. The only remaining edit that was not addressed by the author is the fact that both English and American spellings are used, for example the use of 'randomized' and 'analysed'. Please select whether to use American spelling ('z') or English spelling ('s'), and use it consistently throughout the manuscript.
Author Response
Dear reviewer,
thanks again for the comments to improve the manuscript.
You are right in relation to the relevance of a uniform speeling. In this sense, we finally update the manuscript according to british spelling.
Kind regards.